# Exploring machine learning algorithms in sickle cell disease patient data: A systematic review

**Tiago Fernandes Machado**[1]*, **Francisco das Chagas Barros Neto**[2], **Marilda de Souza Gonçalves**[2,3], **Cynara Gomes Barbosa**[2], **Marcos Ennes Barreto**[4]

1 Institute of Computing, Federal University of Bahia (UFBA), Salvador, Bahia, Brazil, 2 Faculty of Pharmacy, Federal University of Bahia (UFBA), Salvador, Bahia, Brazil, 3 Gonçalo Moniz Institute, Oswaldo Cruz Foundation (FIOCRUZ), Salvador, Bahia, Brazil, 4 Department of Statistics, London School of Economics and Political Science (LSE), London, United Kingdom

* tfernandesm@gmail.com

**Data Availability Statement:** All relevant data are within the paper and its Supporting information files.

## Abstract

This systematic review explores the application of machine learning (ML) algorithms in sickle cell disease (SCD), focusing on diagnosis and several clinical characteristics, such as early detection of organ failure, identification of drug dosage, and classification of pain intensity. A comprehensive analysis of recent studies reveals promising results in using ML techniques for diagnosing and monitoring SCD. The review covers various ML algorithms, including Multilayer Perceptron, Support Vector Machine, Random Forest, Logistic Regression, Long short-term memory, Extreme Learning Machines, Convolutional Neural Networks, and Transfer Learning methods. Despite significant advances, challenges such as limited dataset sizes, interpretability concerns, and risks of overfitting are identified in studies. Future research directions entail addressing these limitations by harnessing larger and more representative datasets, enhancing model interpretability, and exploring advanced ML techniques like deep learning. Overall, this review underscores the transformative potential of ML in increasing the diagnosis, monitoring and define prognosis of sickle cell disease while also highlighting the need for further investigation in the field.

## Introduction

Sickle cell disease (SCD) is an inherited genetic disease that affects hemoglobin, the protein responsible for transporting oxygen in red blood cells [1]. In this disease, red blood cells take on a sickle shape when they lack oxygen, resulting in blockage of blood flow and leading to several clinical manifestations, such as vaso-occlusion, chronic pain, and chronic organ failure [2].

The worldwide prevalence of hemoglobin disorders is estimated to vary between 5% and 6% of the population, with sickle cell disease being the most common among these conditions. Every year, more than 0.3 million individuals are born with some form of hemoglobin disorder, and sickle cell disease is most prevalent in individuals whose ancestors come from sub-Saharan Africa, India, Saudi Arabia, and Mediterranean countries [3, 4].

**Funding:** The author(s) received no specific funding for this work.

**Competing interests:** The authors have declared that no competing interests exist.

Early diagnosis of the disease is of crucial importance to significantly reduce patient morbidity and mortality. Several clinical tests capable of indicating the presence of hematological disease, such as complete blood count, sickle cell test, and hemoglobin electrophoresis, are available to evaluate the morphology of red blood cells and hemoglobin profile. However, these procedures require specialized professionals and laboratory systems being particularly challenging in regions with limited clinical infrastructure [5].

Although science has not yet identified a cure for sickle cell disease, treating its symptoms and relieving pain involves using medications such as hydroxyurea, bone marrow transplantation, and gene therapy [6].

Studies are currently being carried out in hematology using machine learning to diagnose and monitor sickle cell disease. Researchers use algorithms as an alternative to conventional clinical procedures requiring infrastructure and trained experts for data analysis and decision-making.

In this context, we systematically reviewed the literature covering the last five years (2019–2024) to analyze the use of machine learning techniques in examining laboratory, clinical-epidemiological data, and microscopic images of blood smears to support diagnosis and monitoring and its results.

## Materials and methods

### Eligibility criteria

Following the Preferred Reporting Items for Systematic Reviews and Meta-Analyses (PRISMA) recommendations, we conducted a systematic review. Two authors independently searched MEDLINE (via PubMed), Scopus, Web of Science, and IEEE for studies published from January 2019 to April 2024. The inclusion criteria focused on studies investigating the use of supervised, unsupervised, and reinforcement learning models in diagnosing, predicting, monitoring to sickle cell disease. The initial search yielded 279 studies, beginning the preliminary screening phase based on titles and abstracts. Subsequently, we strictly applied inclusion and exclusion criteria, selecting 29 articles for detailed review, as illustrated in Fig 1.

We determined article eligibility based on the following criteria: (1) originality, defined as studies utilizing primary data or available databases, rather than literature reviews, (2) full-article status, (3) results presentation, excluding studies that did not present quantitative or qualitative results related to sickle cell disease, and (4) peer review. Additionally, the studies had to utilize sickle cell disease as their data source and involve classification, regression, or clustering models for diagnostic and/or disease monitoring purposes.

Two different authors individually assessed each article. In cases of eligibility discrepancies, a third reviewer analyzed the article in question and made the final determination regarding its inclusion or exclusion.

### Database selection

We selected four databases based on their relevance to the review topic. We chose Scopus for its extensive coverage as the largest database of abstracts and citations of peer-reviewed literature. We selected PubMed due to its specific focus on health-related outcomes. We opted for Web of Science because it provides comprehensive and complementary information. Additionally, we included IEEE to access specialized results in the computing area. This selection of databases aims to ensure broad and comprehensive coverage of studies related to sickle cell disease.

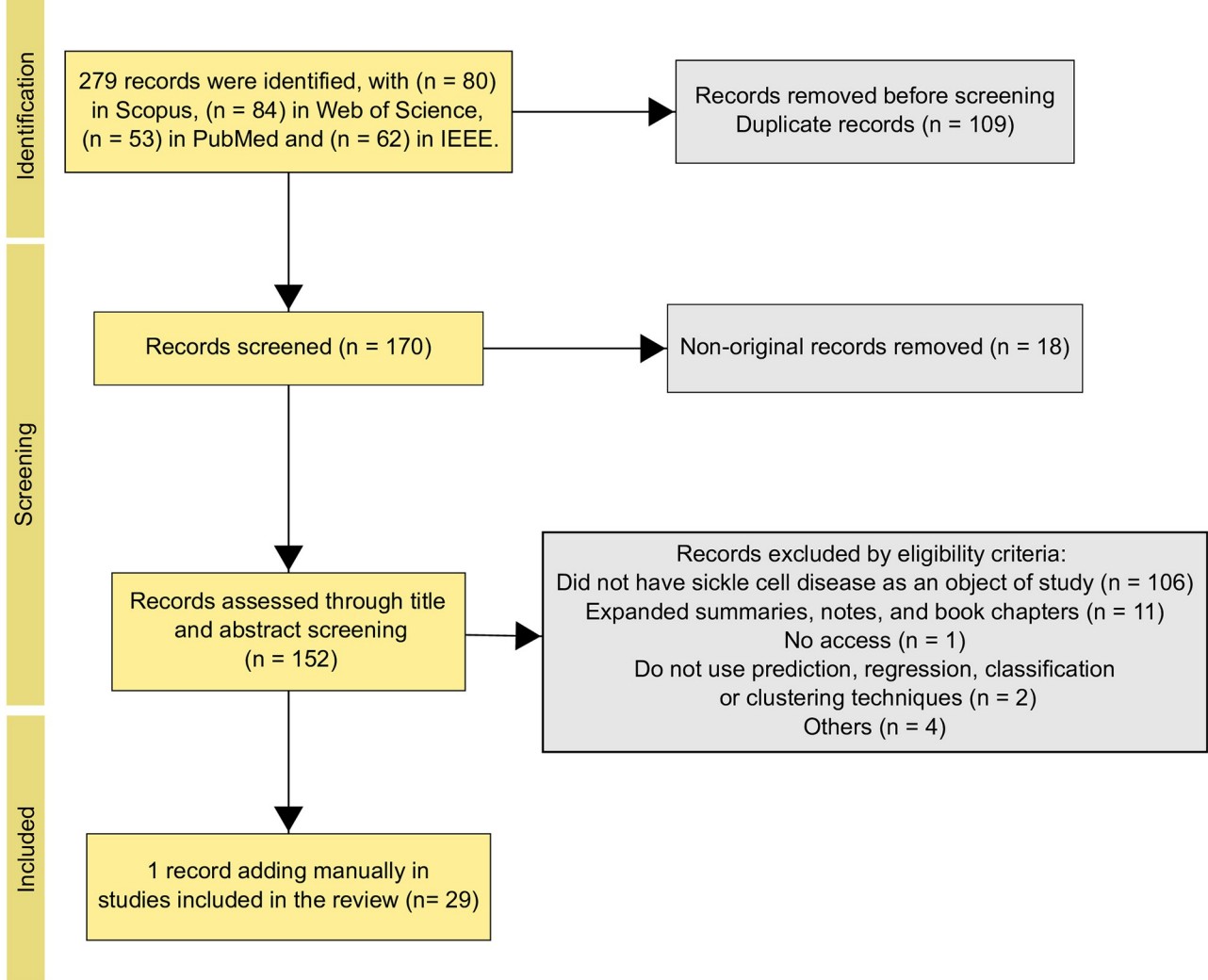

**Fig 1. Preferred Reporting Items for Systematic Reviews and Meta-Analyses (PRISMA) flowchart depicting the studies included in the systematic review.**

## Search strategy

The selected search strategy aimed to optimize the retrieval of pertinent literature by employing broad terms from both thematic areas. For instance, in Scopus, the query utilized the string: TITLE-ABS-KEY("sickle cell disease" OR "anemia sickle cell") AND TITLE-ABS-KEY ("artificial intelligence" OR "machine learning") AND PUBYEAR > 2018, resulting in 80 findings. In Web of Science, the search string was ALL = ((sickle cell disease OR sickle cell anemia) AND (artificial intelligence OR machine learning)), with publication years filtered from 2019 to 2023, yielding 84 results. In the PubMed database, the search string was ("sickle cell disease" OR "sickle cell anemia") AND ("artificial intelligence" OR "machine learning"), also with a five-year filter, and it retrieved 53 studies. In IEEE, the search string ("sickle cell disease" OR "sickle cell anemia") AND ("artificial intelligence" OR "machine learning"), with the same temporal filter, led to the identification of 62 relevant works. The exact syntax used for each database to search for records can be found in S2 File.

## Information extraction strategy

To extract information from the studies, we adopted the PICO (Patient, Intervention, Comparison, Outcome) strategy from evidence-based medicine. This strategy involves formulating questions to obtain guiding information. It aids in searching and selecting relevant studies for the review. We extracted features for analysis from each selected study, including the studied population, the type of data input used, the use of synthetic data (if applicable), the classifier used for diagnosis, the performance metric used, and the part of the model to which researchers applied it (either training or validation). These pieces of information are essential for assessing the approach and results of the selected studies.

## Study risk of bias assessment

We conducted a life-risk analysis on 29 selected articles that focus on applying machine learning techniques to diagnosing and monitoring sickle cell disease. Assessing the risk of bias is an important component of systematic reviews, ensuring the reliability and validity of the included studies. Our analysis focuses on four main areas: (a) Participant Selection and Data: we evaluate the representativeness and selection criteria of participant data to identify selection bias; (b) Data pre-processing: we evaluate the suitability of pre-processing techniques, including class balancing, normalization, and missing data imputation; (c) Algorithm performance: we evaluate the robustness and generalization of the machine learning algorithms used, including the use of cross-validation and independent testing of datasets; (d) Results Report: we evaluate the statistics presented in the results reports regarding their comprehensiveness and transparency, to observe distortions in the metrics. The risk of bias was carried out independently by two authors. In cases where there were disagreements in the assessment, a third author reviewed the analysis and made the final decision regarding the risk of bias. This process ensures that the evaluation is impartial, objective, and as accurate as possible, contributing to the quality of the systematic review (see S3 File).

## Results

Of the 29 included studies, 15 focused on sickle cell disease (Table 1), and 14 on disease monitoring (Table 2). This monitoring included risk prediction for mortality, acute organ failure, identification of pain crises, assessment of the effects and dosage of hydroxyurea, as well as summarizing treatment approaches and hospital discharge.

Fig 1 describes the search and selection process, detailing the number of records identified and the number of studies selected for review. A list of all articles returned is in S2 File. The risk of bias assessment for each study included in the systematic review is available in S3 File. The selected articles were classified in two categories: diagnostic and monitoring studies.

## Machine learning approaches for diagnosis

Significant strides have been made in harnessing the potential of machine learning algorithms to diagnose sickle cell disease using images of blood smears and hematology data. These studies have been instrumental in employing computer vision techniques and classification algorithms to analyze the morphological characteristics of blood cells in smear images. Several variables, including shape, size, texture, and other pertinent cellular properties, have been meticulously considered in the analyses. Based on this information, researchers have successfully trained algorithms to identify and differentiate normal human cells affected by sickle cell disease. Table 1 summarizes the main research findings and contributions.

**Table 1. Diagnostic studies based on machine learning methods.**

| Study | Method/Data(sample size) | Intervention or Paradigm | Main Results |
|---|---|---|---|
| [3] | Supervised Learning / Blood smear images (80) | Compared three classification algorithms: K Nearest Neighbor, Support Vector Machine and Extreme Learning Machine, to detect the presence of sickle cell anemia or not. | The ELM algorithm presented accuracy of 95.4%, sensitivity of 87.5%, specificity of 83.3%, accuracy of 87.7% and f-score of 91.3%. |
| [4] | Supervised Learning / Blood smear images (2695) | Consider clusters of cells and cells at the edge of the image for classification using Random Forest, Extra Trees and Gradient Boosting. | The GB algorithm presented SDS-score of 94.6%, F-measure of 94.6%, Class Balance Accuracy (CBA) of 93.9%, and Mathews Correlation Coefficient (MCC) of 88.7%. |
| [5] | Deep Learning / Blood smear images (96) | They used a smartphone microscope to obtain the blood smear images and classified the images using Deep Learning to discover patients without and with the presence of sickle. | Accuracy of 98% and Area under the ROC curve of 0.99. |
| [7] | Deep Learning / Blood smear images (629) | Classify blood smear images into three classes: circular (normal), elongated (sickle cells) and others with a Convolutional Neural Network. | Accuracy of 94.5% with a CNN composed of 5 convolution layers. |
| [8] | Supervised Learning / Clinical data (1387) | Diagnoses of sickle cell, normal cells and thalassemia with the aid of the Multilayer Perceptron classifier. | Accuracy of 96% for the test set, and in a new validation set of 100 samples, they obtained a result with 99.0% accuracy. |
| [9] | Deep Learning / Blood smear images (1815) | Used machine learning to classify microscope images into 5 categories: normal cells, malaria, sickle cell anemia, megaloblastic anemia or thalassemia with CNN. | The accuracy was 93.4%. |
| [10] | Deep Learning / Blood smear images (527) | Tested the classification of blood smear images into the three classes with various CNN algorithms. | Using the Inception V3 algorithm, they obtained an accuracy of 91.0%. |
| [11] | Transfer Learning / Blood smear images (90) | They addressed AlexNet, which is a pre-trained CNN model for classifying disease cases versus traits. | Accuracy of 95.5%, sensitivity of 94%, and precision of 95%. |
| [12] | Supervised Learning / Clinical data (1428) | To identify the best algorithm for detecting the disease, for this the authors compared five classifiers: Naive Bayes, Neural Networks (NN), Support Vector Machine, J48 and PART. | Accuracy of 92.2% with a time of (0.01s) for the Naive Bayes classifier. |
| [6] | Supervised Learning / Blood smear images (1000) | They used Support Vector Machine, Naive Bayes classifiers and logistic regression machine learning classifiers to tell whether a blood sample with overlapping blood cells was normal or presented some pathology. | The SVM algorithm showed better results with accuracy, sensitivity and specificity of 98.1%, 98.2%, and 97.9%, respectively. |
| [13] | Transfer Learning / Blood smear images (10002) | They applied transfer learning models such as ResNet-50, AlexNet, MobileNet, VGG-16, and VGG-19, and convolutional neural network to classify sickle cell disease into circular, elongated, and others. | The ResNet-50 algorithm achieved 100% precision, recall, and F1 score for circular, elongated, and other cell shapes. |
| [14] | Supervised Learning / Blood smear images (624) | They proposed a set of hybrid classification models to diagnose sickle cell disease. | RF and Extreme Gradient Boosting Algorithm (XGBoost) called RF-XGBoos showed 99% accuracy for the dataset. |
| [15] | Transfer Learning / Blood smear images (1985) | They used pre-trained deep convolutional neural networks (CNNs) models for sickle cell and normal cell classification tasks. | The ResNet-50 algorithm showed an accuracy of 93.8%. |
| [16] | Deep Learning / Blood smear images (340) | Classification with Recurrent Neural Networks (RNN) to differentiate normal from abnormal sickle cells. | Accuracy, precision, recall and F1-score, obtained 99.8%, 99.7%, 98.4%, and 98.5% respectively. |
| [17] | Transfer Learning / Blood smear images (1664) | Present comparative results of three models: GoogLeNet, ResNet-18, and ResNet-50 for detecting sickle cell samples; the authors used different hyperparameters and introduced Explainable Artificial Intelligence (XAI) through Grad-CAM to make the predictions transparent and understandable. | ResNet-50 showed the highest accuracy of 94.9%. |

Studies have addressed various data sets to investigate sickle cell disease diagnostic methods, [4, 7, 10, 13, 14] explored the erythrocytesIDB database, using both the original data and, in some cases, synthetic data to train their classification models. Already [5] obtained images using a specific smartphone, while [8, 11, 15–17] chose data from hospitals and research institutes. The work of [12] gathered clinical information from patients diagnosed with sickle cell disease over 12 years in a university hospital, and [6] used blood smear images from the

**Table 2. Monitoring studies based on machine learning methods.**

| Study | Method/Data(sample size) | Intervention or Paradigm | Main Results |
|---|---|---|---|
| [18] | Supervised Learning / Photoplethysmography (212) | Predict an increased propensity for pain attacks during sleep. | The multi-layer perceptron had a performance of F1-score 0.42, Precision 0.29, Recall (sensitivity) 0.79, Specificity 0.60, and Accuracy 0.63. |
| [19] | Supervised Learning / Vital signs (59728) | Pain intensity. | The best algorithm for the problem was the Decision Tree, working to predict pain on a severity scale (from 0 to 10), with an accuracy of 0.72 at an inter-individual level (between patients) and 0.65 at an intraindividual (within each patient). |
| [20] | Supervised Learning / Vital signs (33000) | Pain intensity. | Accuracy of 82.3%. |
| [21] | Supervised Learning / Clinical data (2395) | They tested three different machine learning classification models to identify the pain index for VOC during admission and after discharge. | They pointed out the best Random Forest model, with a micro-average accuracy of 92%, micro-average F1 score of 0.63, mean squared error of 1.1, and area under the receiver operating characteristic curve of 0.90, highlighting its predictive performance for the highest pain ratings greater than six on a scale of (0–10). |
| [22] | Supervised Learning / Clinical data (1896) | They proposed using machine learning techniques to identify the dosage of hydroxyurea or hydroxycarbamide, which they divided into nine categories. | The Levenberg-Marquardt Neural Network algorithm obtained the best results on the data, presenting an accuracy of 0.84 and AUC of 0.87 in identifying the correct dosage of hydroxyurea or hydroxycarbamide in (mg). |
| [23] | Supervised Learning / Clinical data (1128) | Dosage of hydroxyurea. | For the Long short-term memory algorithm, the accuracy was 78%. |
| [24] | Supervised Learning / Clinical data (304) | Effect of hydroxyurea. | For the Support Vector Machine algorithm, the accuracy was 94%, 95% sensitivity, and 91% specificity. |
| [25] | Unsupervised Learning / Clinical data (125) | Admission to intensive care unit (ICU). | As a result, the authors claimed to have successfully found three SCD clusters capable of clearly identifying the most severe group (cluster 2), an intermediate severe group (cluster 1), and a less severe group (cluster 3). |
| [26] | Supervised Learning / Clinical data (446) | Hospital readmission. | LR (C-statistic 0.77, IC 95% 0.73–0.80). |
| [27] | Supervised Learning / Clinical data (191406) | Reasons to predict hospital discharge class. | The J48 and Naive Bayes algorithms had better classification results, with accuracy of 95% and 92%. |
| [28] | Supervised Learning / Clinical data (404) | Decline in kidney function. | The estimated decline in GFR $> 3$ mL/min/1.73 $m^2$ at 6 months was predicted with 82% sensitivity, 80% accuracy, and AUC of 0.88 (95% CI: 0.79–0.97), while eGFR decline $> 5$ mL/min/ 1.73 $m^2$ at 6 months was predicted with sensitivity of 79%, accuracy of 84%, and AUC of 0.91 (95% CI: 0.85– 0.97). |
| [29] | Supervised Learning / Physiological characteristics (63) | Organ failure. | The Multilayer Perceptron algorithm was able to predict organ failure up to 6 hours before onset, with averages of 96% sensitivity and 98% specificity. |
| [30] | Supervised Learning / Clinical data (63) | They studied the prediction of the onset and severity of comorbidities in patients with sickle cell disease. | They revealed that Random Forest was the most accurate method, with 88% precision, 82% recall, and 92% specificity in predicting disease severity in sickle cell patients. |
| [31] | Supervised Learning / Clinical data (600) | Risk score for mortality prediction. | The proposed model stratified patients into four groups with significantly different 4-year mortality rates (3%, 11%, 35%, and 75%, respectively). |

Hematological Atlas digital library to detect blood cell overlaps. The study of [9] used data available on the Kaggle platform.

The studies employed image-processing techniques to improve the analysis of blood smears. Some like [3, 7] chose to convert the images to grayscale, while [16] focused on noise filtering to improve image quality. The applied advanced segmentation techniques to separate blood cells from the background used by [4, 6]. In the approaches of [11, 15, 17] they used feature reduction and data augmentation techniques to improve the robustness of their machine-learning models. Among the methodologies adopted, segmentation based on Otsu, the Wiener

filter, and image optimization techniques such as Whale Particle Swarm Optimization, as mentioned by [16], updating the classification correction. Furthermore, the most requested scenarios, such as the detection of overlapping cells and the presence of irrelevant artifacts in blood smear images, were also tackled by researchers.

The studies explored a set of classification algorithms, including k-nearest neighbors (KNN), Support Vector Machine (SVM), and Extreme Learning Machine (ELM), compared by [3, 6]. Used Random Forest (RF), Extra Trees (ET), and Gradient Boosting (GB) [4], while Convolutional Neural Networks (CNN) were evaluated by [5, 7, 9, 11, 16]. Explored Multilayer Perceptron (MLP) [8], used ResNet, VGG, and Inception [10, 13, 15, 17]. Investigated Naive Bayes, J48, and PART [12], while [14] applied hybrid models.

To evaluate the performance of the models, researchers used metrics such as precision, sensitivity, specificity, accuracy, F-score, and area under the ROC (Receiver Operating Characteristic) curve. Studies that employed deep learning and transfer learning, mainly using architectures like ResNet-50, consistently reported the highest accuracies, with some achieving up to 100% on specific metrics. These findings demonstrate that deep learning-based models, especially CNNs and their pre-trained variants like ResNet-50 and Inception V3, outperformed other approaches regarding accuracy and other evaluation metrics for diagnosing sickle cell disease. Reported accuracies varied widely, ranging from 73.33% with KNN to 99.8% with hybrid RNN models and optimization techniques, highlighting the superior performance of advanced deep learning methods.

The approach taken by [8] differs from other studies using images of blood smears. They explored the use of a Multilayer Perceptron with clinical patient data, using 13 parameters from the dataset: age, hemoglobin, hematocrit, red blood cell distribution width, mean corpuscular volume, mean corpuscular hemoglobin, mean hemoglobin concentration corpuscular, red blood cells, reticulocytes, fetal hemoglobin, HBA0, HBA2, and the diagnostic label. The authors claimed an accuracy of 96.04% on the test set. Furthermore, when validating the model with a new set of 100 samples, achieved an accuracy of 99%.

Even without imaging results, traditional algorithms like MLP can achieve performance comparable to deep learning-based models when well-tuned and fed with robust clinical data. Fig 2 summarizes the main aspects and methodologies researchers adopt, providing an overview of the elements used in diagnosing sickle cell disease in the studies consulted.

Machine learning has shown significant success in diagnosing SCD using blood smear images and hematology data, with several approaches achieving high levels of accuracy. Studies indicate that classification techniques, especially convolutional neural networks, have excelled in analyzing morphological features, with some models reaching up to 100% accuracy on specific metrics. However, automated SCD diagnosis remains an unresolved challenge. Problems such as noise and irrelevant artifacts in blood smear images, difficulties in detecting overlapping cells, and the generalization of models persist, as many studies rely on specific or synthetic databases, limiting the applicability of classifiers to broader clinical scenarios and populations. While CNNs and their pre-trained variants have outperformed simpler algorithms in accuracy, there is still room for exploring and comparing new models.

## Machine learning approaches for monitoring

Researchers have expanded the possibilities for applying classifiers in monitoring, resulting in various studies exploring the use of these algorithms in areas such as medication dosage, pain classification, and outcome prediction. Table 2 summarizes the main results found in the last five years, addressing these aspects.

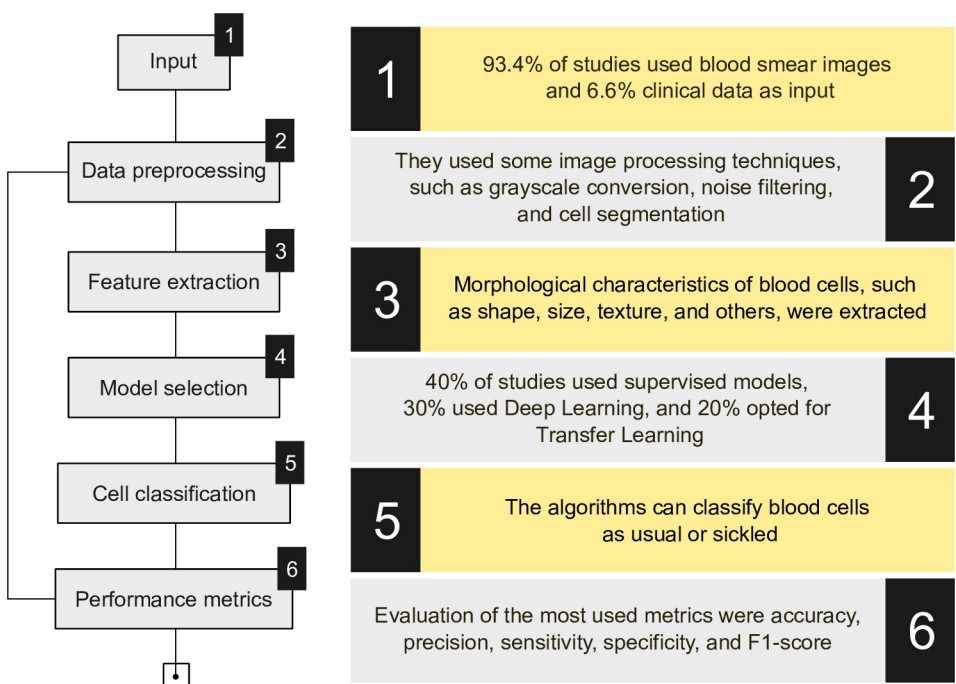

**Fig 2. Summary of the main aspects and methodologies adopted by researchers.**

In an attempt to use photoplethysmography (PPG) data collected during the sleep of sickle cell disease patients to predict an increased propensity for pain crises, [18] adopted a machine learning approach using a dataset derived from the multicenter Sleep and Asthma Cohort (SAC) study, involving children and adolescents. The SAC dataset consisted of 212 individuals, with 36 belonging to the positive category ("high pain") and 176 belonging to the negative category ("low pain"). The authors employed a two-level stacked machine learning approach, incorporating detailed features extracted from the PPG signals. The authors used several machine learning algorithms at the first level to predict each individual's pain category based on PPG characteristics and other clinical information. In the second level, a metamodel utilized inputs from the first-level model and the model's outputs to produce the final prediction. Among the algorithms used in the first level were multiple logistic regression, linear kernel Support Vector Machine, radial basis function (RBF) kernel SVM, random forest, ExtraTrees, AdaBoost, one-dimensional CNN, multilayer perceptron, polynomial kernel SVM, gradient boosting trees, and extreme gradient boosting (XGBoost). The multilayer perceptron substantially outperformed other base models and was selected as the representative algorithm for the "metamodel" in the second level of the proposed stacking model. Evaluation metrics for the model included F1-score (0.4255), precision (0.2915), recall (sensitivity) (0.7917), specificity (0.6048), and accuracy (0.6366). These results indicate the feasibility of developing a wearable system with a higher propensity for more frequent vaso-occlusive crises.

To address patients' subjectivity of pain intensity during hospital visits, [19] proposed an approach using electronic health records. The study collected data from 47 patients over five consecutive years at Duke University Hospital, totaling 59,728 records. Each record contained measurements of six vital signs: peripheral capillary oxygen saturation, systolic blood pressure, diastolic blood pressure, heart rate, respiratory rate, and temperature. In addition to vital signs, each record included patient self-reported pain scores, ranging from 0 (no pain) to 10

(intense and unbearable pain). The authors applied five supervised machine learning classification algorithms—k-nearest Neighbors, Support Vector Machine, Multinomial Logistic Regression (MLR), Decision Tree (DT), and Random Forest—to predict patient pain scores based on vital signs. The metric used to evaluate the algorithms was accuracy. The Decision Tree model showed the best performance in predicting pain on a severity scale of 0 to 10, with an accuracy of 0.728 at the inter-individual level (among patients) and 0.653 at the intraindividual level (within each patient). The accuracy of the DT model significantly improved to 0.941 when reducing the rating scale to two points (i.e., no pain/mild pain: 0–5, intense pain: 6–10) at the individual level.

To enhance the prediction of pain intensity in patients during hospital visits, the authors of [20] delved into data regarding medications administered to sickle cell disease patients. They utilized a database comprising 126519 electronic health record (EHR) entries from 496 SCD patients collected over five consecutive years at Duke University Hospital. However, they could only include 33000 records due to missing information in the medical records. Medical staff documented vital signs every 4 hours during hospitalization while documenting medication data as they administered them to patients, including details such as medication type, status (administered, forgotten, removed, or expired), and total dosage (mg/mL). Machine learning techniques, specifically deep autoencoder neural networks, were utilized to impute missing data in the medical records. Subsequently, they employed supervised learning techniques like Random Forest, Lasso regression, and Support Vector Machine to predict pain intensity based on available data. They assessed model performance using accuracy as the metric. The results underscored the significance of medication information, with the new model achieving an accuracy of 82.3%, indicating an improvement over the authors' previous model. These findings emphasize the necessity of a broader data cohort and the adoption of deep representational machine learning techniques to enhance the accuracy of predicting pain intensity in patients during hospital visits.

Using a novel approach compared to prior research in gathering input data, [21] remotely monitored 19 patients (74% HbSS genotype) between April and June 2022 for an average of 28 days. They generated a dataset comprising 2395 pain points using wearable technology during the hospitalization of sickle cell disease patients experiencing vaso-occlusive crises (VOC). The researchers instructed participants to report daily pain scores (ranging from 0 to 10) through a mobile application called Nanbar and to wear an Apple Watch continuously for up to 30 days after discharge. Data included heart rate (resting, mean, and variability) and step count. Demographic information, SCD genotype, and hospitalization details, including pain scores reported to nurses, were extracted from electronic medical records. Three machine learning classification models were used to identify pain index during VOC admission and post-discharge. The authors identified the best-performing model as the Random Forest model, with a micro-average accuracy of 92%, micro-average F1 score of 0.63, mean squared error of 1.1, and area under the receiver operating characteristic curve of 0.9, highlighting its predictive performance for pain scores exceeding 6.

In [22], the authors proposed the use of machine learning techniques to identify the dosage of hydroxyurea or hydroxycarbamide discretized into nine categories, covering a range of 250 to 1500mg from a dataset of blood tests from 1896 sickle cell disease patients collected over six years at the Hemophilia Center of Alder Hey Children's Hospital (NHS). A total of 14 features, such as weight, mean corpuscular volume, reticulocyte count, hemoglobin, reticulocyte count, HbF, platelets, lactate dehydrogenase, neutrophils, alanine aminotransferase, bilirubin, aspartate aminotransferase, bio-blood corporal, and medication doses, were used to train four machine learning algorithms: Logistic Regression (LR), Quadratic Discriminant Classifier (QDC), Naïve Bayes Classifier (NBC), and Levenberg-Marquardt Neural Network (LEVNN).

Accuracy and area under the ROC curve were the performance metrics used by the authors to show that the LEVNN classifier achieved the best results on the test data compared to others, with an accuracy of 0.846444 and AUC of 0.871889 in identifying the correct dosage of hydroxyurea or hydroxycarbamide in (mg).

Aiming at monitoring pain episodes in patients with sickle cell disease who receive hydroxyurea (HU) treatment, [23] conducted a study using a dataset consisting of 1128 samples obtained from a hospital in Liverpool, United Kingdom. The study's primary goal was to compare the performance of two classification models in categorizing HU dosages as low, high, and moderate for sickle cell disease patients. The authors assessed 12 parameters as input for the machine learning models. These parameters included patient weight in kilograms, aspartate aminotransferase enzyme activity, alanine aminotransferase enzyme activity, lactate dehydrogenase enzyme activity, mean corpuscular volume, bilirubin levels, hemoglobin levels, fetal hemoglobin (HbF) levels, neutrophil count, reticulocyte count (RETIC A and % RETIC), and platelet count. The study used Long short-term memory (LSTM) and Extreme learning machine models to classify HU dosages. The results showed that the LSTM model achieved an overall accuracy of 78%, while the ELM model achieved an accuracy of 52%.

Trying a different approach [24], researchers investigated the impact of hydroxyurea on 304 sickle cell disease patients using predictive machine-learning techniques. The classification criteria relied on measuring fetal hemoglobin levels before and after hydroxyurea treatment. Patients who experienced a 15% or more increase in Hbf value post-treatment were categorized as class 1 (responders), while those with less than a 15% increase were classified as class 0 (non-responders). The study involved 18 parameters characterizing each patient, including Age, Sex, HbF, HbA2, HbS, Hematocrit, Hb, MCV, MCH, RDW, TLC, DLC (L, M, G), Plc, EnzSGPT, and EnzSGOT. These parameters served as inputs for the Support Vector Machine algorithm, achieving an accuracy of 94%, sensitivity of 95%, and specificity of 91%.

The study conducted by [25] aimed to analyze the admissions profile in intensive care units (ICUs) for sickle cell disease patients. To do this, the authors examined 125 patient records obtained from admissions between 1996 and 2020 at the ICU of Hospital das Clínicas in São Paulo. Using unsupervised machine learning clustering techniques, the authors grouped patients based on characteristics without prior hypotheses to identify groups related to severity, prognosis, or the level of care required. These groupings are essential for improving patient care and assisting the healthcare system in allocating ICU beds for the most severe cases. From the silhouette plot, the optimal number of clusters identified was three.

Cluster 1 was the largest (n = 69), with the highest proportion of the SS genotype (87%). This cluster had the lowest prevalence of comorbidities such as heart failure, chronic kidney disease, and hepatitis C virus but a high proportion of patients with a history of stroke. Upon admission to the ICU, this cluster had the highest prevalence of reasons related to sickle cell disease (72.5%) and liver dysfunction (76.8%). A low proportion of patients required invasive mechanical ventilation (11.6%), and in-hospital mortality was 18.8.

Cluster 2 (n = 25) was the most severe and had older patients (average age of 37 years). Most were male (76%) with a high prevalence of the SS genotype (80%). This cluster had the highest prevalence of hypertension and chronic kidney disease. Upon ICU admission, this cluster also had the highest severity scores, along with a high prevalence of kidney and neurological dysfunction. Only 3 (12%) of the patients were admitted for reasons related to sickle cell disease. Most of the mechanically ventilated patients were in this cluster, and hospital mortality was also high (44%). Out of the 11 deaths in Cluster 2, eight patients were on mechanical ventilation.

Finally, Cluster 3 (n = 31) mainly consisted of women (96.8%) with heterozygous S/$\beta$ genotype (64.5%). Among these patients, 19 (61.3%) had admissions related to sickle cell disease.

The most common comorbidities in this cluster were heart failure (45.2%), stroke (35.5%), and chronic hepatitis C (41.9%). Mortality was the lowest (12.9%). As a result, the authors successfully identified three sickle cell disease clusters capable of clearly distinguishing the most severe group (Cluster 2), an intermediate group (Cluster 1), and a less severe group (Cluster 3).

In a retrospective study using electronic health records, [26] investigated the ability of machine learning algorithms (ML) to surpass standard hospital readmission scoring systems, such as the LACE and HOSPITAL indices. The study included 446 adult patients with sickle cell disease and 3299 unplanned hospital admissions. Three ML algorithms, Logistic Regression, Support Vector Machine, and Random Forest, were applied and evaluated by the authors. The considered features included laboratory data, demographic data, the number of outpatient visits before the current admission, and the number of Emergency Department (ED) visits before the current admission. Records from five University of Pittsburgh Medical Center (UPMC) hospital system hospitals provided the data. The results obtained from the dataset showed that the ML algorithms outperformed the LACE (C-statistic 0.6, 95% Confidence Interval—CI 0.57–0.64) and HOSPITAL (C-statistic 0.69, 95% CI 0.66–0.72) indices. The Random Forest algorithm exhibited a C-statistic of 0.77 (95% CI 0.73–0.79), while Logistic Regression achieved a C-statistic of 0.77 (95% CI 0.73–0.8), both demonstrating superior performance.

In a detailed analysis, [27] examined 191,406 medical records from 1428 patients diagnosed with sickle cell disease at King Fahad University Hospital in eastern Saudi Arabia. The dataset encompasses the years 2008 to 2020. The objective was to employ algorithms based on hospitalization records, visits, and other factors to predict the discharge class (normal routine discharge, against medical advice, transfer to another hospital, expiration or death, and escape from the hospital) for various types of patients with sickle cell disease. The study employed five classifiers—Naive Bayes, J48, SVM, NN, and PART—and compared their accuracy, kappa statistics, and classification time. The findings indicated that J48 and Naive Bayes demonstrated superior classification results, achieving accuracies of 95% and 92%, respectively, along with shorter processing times compared to other classifiers.

In their study, [28] delved into the potential of machine learning to predict rapid renal function decline in sickle cell disease patients. They hypothesized that the models could forecast rapid renal function decline in severe SCD genotypes six months to 1 year in advance. Their dataset comprised an internal cohort with 236 patients and an external cohort with 168 patients (HbSS). Input data for the algorithms encompassed demographic information, laboratory test results, SCD-related complications, comorbidities, and treatments. The glomerular filtration rate (GFR) was estimated at each visit using the CKD-EPI equation (Chronic Kidney Disease Epidemiology Collaboration). They obtained results indicating that they predicted the estimated GFR decline of $>3$ mL/min/1.73 m$^2$ in 6 months with a sensitivity of 82%, accuracy of 80%, and an AUC of 0.88 (95% confidence interval [CI]: 0.79–0.97). Additionally, they predicted the decline in eGFR of $>5$ mL/min/1.73 m$^2$ in 6 months with a sensitivity of 79%, precision of 84%, and an AUC of 0.91 (95% CI: 0.85–0.97). Regarding the estimated GFR decline in 12 months, sensitivity stood at 53%, accuracy at 70%, and the AUC at 0.67 (95% CI: 0.53–0.81) with the threshold of $> 3$ mL/min/1.73 m$^2$, while sensitivity was 64%, precision was 65%, and the AUC was 0.77 (95% CI: 0.67–0.87) with the threshold of $> 5$ mL/min/1.73 m$^2$ for patients in the internal cohort. The authors concluded that further studies are necessary to assess the utilization of machine learning algorithms in SCD-related kidney disease.

Thinking about early detection of organ failure in adult patients with sickle cell disease [29]. They conducted the research in the intensive care units (ICU) of Methodist Le Bonheur Healthcare hospitals located in Memphis, Tennessee, United States, with the participation of

63 patients and the analysis of 163 events. During the study, the researchers considered five physiological characteristics of the patients: heart rate, respiratory rate, systolic blood pressure, diastolic blood pressure, and mean arterial pressure. These data served as input for four machine learning algorithms: Multilayer Perceptron, Support Vector Machine, Random Forest, and logistic regression methods. The researchers used sensitivity and specificity metrics to evaluate the algorithms. MLP demonstrated promising results, being able to predict organ failure with an advance notice of up to 6 hours before the onset. MLP showed an average sensitivity of 96% and specificity of 98%, indicating its effectiveness in the early detection of this critical clinical outcome.

In [30], the authors employed advanced techniques in health informatics and machine learning to analyze hemoglobin S molecules in sickle cell disease patients, aiming to predict the onset and severity of comorbidities. The researchers categorized patients into three severity classes of the disease: non-severe (0), severe (1), and highly severe (2). They used input data from 63 sickle cell anemia patients sourced from the Sickle Cell Institute in Chhattisgarh, Raipur. This data included hematological parameters such as hemoglobin, red blood cells, and hematocrit. The authors utilized spectroscopy to analyze the spectral properties of hemoglobin S molecules, which play a crucial role in red blood cell sickling. They compared two machine learning classifiers with different kernels: Random Forest, Linear Support Vector Machine, Radial Support Vector Machine, Polynomial Support Vector Machine, and Sigmoid Support Vector Machine. The study revealed that Random Forest was the most accurate method, achieving 88% precision, 82% recall, and 92% specificity in predicting disease severity in sickle cell patients. These results highlight the effectiveness of Random Forest in handling sickle cell disease patient data.

In its most serious outcome, [31] proposed a new risk score combining clinical, laboratory, and imaging data to predict mortality in sickle cell disease patients. The study analyzed 600 adult patients with this condition, and researchers obtained the data from patients enrolled at the National Heart, Lung, and Blood Institute (NHLBI). The features used in the study included tricuspid regurgitant velocity, estimated proper atrial pressure, mitral E velocity, left ventricular septal thickness, body mass index, blood urea nitrogen, alkaline phosphatase, heart rate, and patient age. To handle missing values in the covariates, the authors applied random survival forests (RSF) as an imputation method and to provide variable ranking based on predictive power. Subsequently, the authors used regularized Cox proportional hazards regression to analyze the most critical predictors RSF selected to develop the best predictive model and risk score. The results demonstrated that the proposed prognostic risk score outperformed, with a bias-corrected C statistic of 0.763. Additionally, the model could stratify patients into four groups with significantly different 4-year mortality rates, 3%, 11%, 35%, and 75%, respectively. This stratification highlighted the risk score's ability to identify distinct risk profiles among SCD patients and provide relevant prognostic information for clinical decision-making.

Fourteen articles cover various topics: four address pain crises, three discuss hydroxyurea manipulation, three focus on hospitalization events, one examines the decline in kidney function, one investigates comorbidities, one looks into organ failure, and one explores mortality (see Fig 3). Among these, only one utilized unsupervised machine learning clustering techniques [25]. The selected studies present some findings: all input data came from medical or electronic records. Researchers chose traditional supervised algorithms to classify sickle cell disease outcomes in these cases. The commonly used performance metrics were accuracy, sensitivity, specificity, precision, and F1-score.

The heterogeneity of outcomes and symptoms in patients with SCD makes clinical monitoring a complex task. Machine learning algorithms have demonstrated their potential in

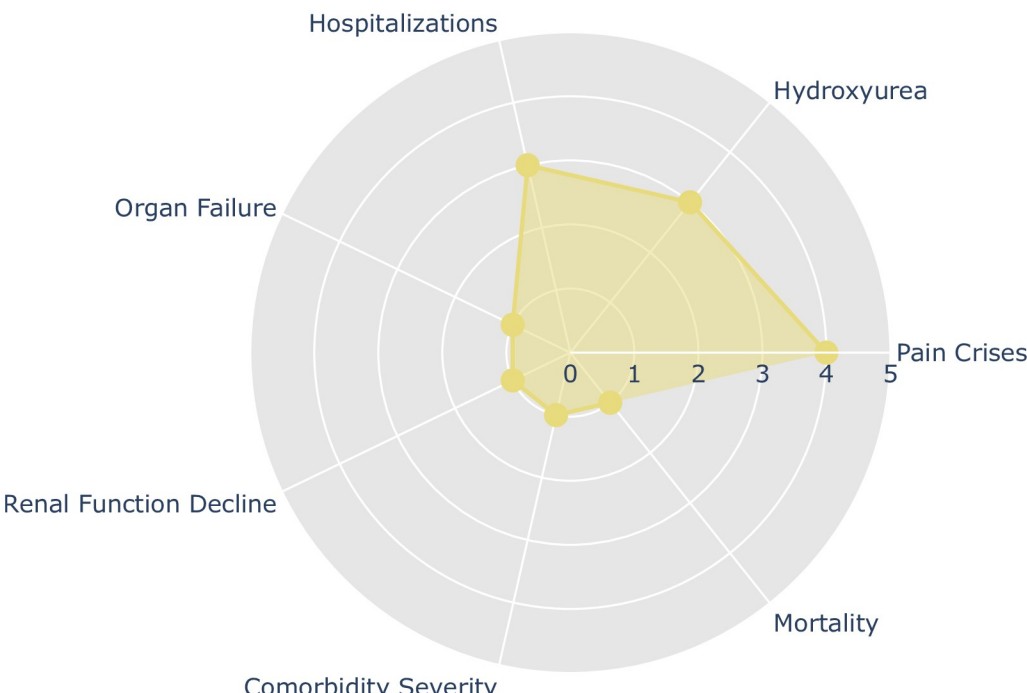

**Fig 3. Number of monitoring studies selected in the last five years according to their monitoring category.**

various applications, ranging from pain assessment to predicting organ failure. However, the challenge of monitoring remains unresolved. Key issues include the difficulty in generalizing across diverse disease outcomes, the limited size of available databases, the use of synthetic data, identifying the most influential biomarkers for specific clinical conditions, and managing the occurrence of multiple simultaneous outcomes. These represent future challenges for researchers in the field.

## Discussion

Our results prove that machine learning techniques have a high potential in detecting and monitoring sickle cell disease. CNNs and transfer learning techniques performed better in diagnosing sickle cell disease than algorithms such as support vector machines, random trees, K-nearest neighbors, and logistic regression. They achieved more significant accuracies than 90% and provided superior generalization ability in blood smear image classification tasks. These findings are consistent with previous studies demonstrating the effectiveness of machine learning models in analyzing biomedical data. For example, [5, 7, 9, 11, 16] also reported high accuracy when using CNNs for sickle cell disease classification, corroborating our results.

Several studies included in the review present some specificities: [3, 5, 11, 29, 30] used relatively small samples, which may limit the generalization of the results. Furthermore, considerable variations in preprocessing methodologies, stage labeling, and learning algorithms were used, making direct comparisons between manuscripts difficult. Some researchers reported the use of cross-validation methods [4, 12, 18–21, 24, 27, 29, 30], however, the vast majority did not report this methodology in their manuscripts, which may introduce bias into the results. Furthermore, no external validation was applied, which may compromise the reproducibility results in different populations.

Despite the methodological rigor of a systematic review, the search strategy cannot always capture all relevant studies, especially those published in languages other than English. Furthermore, although two independent reviewers conducted the risk of bias assessment, subjectivity in the process may have influenced the results. The lack of consensus in some cases resulted in the intervention of a third reviewer, which, although necessary, could introduce additional variations in the evaluation.

Future research should focus on studies with more extensive and more diverse samples and explore new deep and hybrid learning approaches. Standardization of validation methods and transparency in reporting results are also essential to facilitate the comparison and reproduction of studies.

In summary, this systematic review provides a comprehensive overview of the use of machine learning techniques in detecting and monitoring sickle cell disease. Despite the limitations of the included studies and the review process, the results highlight the significant potential of these technologies. The practical application of these techniques could transform the management of sickle cell disease, and future research should continue to explore and refine these approaches to maximize their clinical impact.

## Conclusion

In this work, by analyzing the studies presented in the systematic review on the use of machine learning algorithms in detecting and monitoring sickle cell disease from various input data, we can identify several significant findings and some trends and areas that can be explored in future work.

A significant discovery is the effectiveness of machine learning algorithms in classifying normal and sickle cell disease-affected blood cells. Different techniques, from classical algorithms such as Randon Forest, k-nearest neighbors, and support vector machine to convolutional neural networks, have been successfully applied for this purpose. These algorithms have achieved high levels of accuracy, sensitivity, and specificity in disease detection, and they have the potential to be auxiliary diagnostic tools.

Furthermore, some innovative approaches have been explored, such as the use of Transfer Learning, to improve the performance of classification model studies further. The transfer of knowledge from pre-trained models proved to be especially effective, allowing the adaptation of deep neural networks to classify comorbidities presented by the disease condition.

However, some challenges still need to be addressed. One of them is the availability of representative and well-annotated datasets. Many studies are based on limited data sets, which can compromise the ability of models to generalize results to different settings and clinical conditions. This lack of representation in the data can lead to overfitting of models, where they have adapted to the training data, resulting in reduced performance when applied to new data.

Furthermore, the interpretability of machine learning models is a growing concern, especially in medical applications. Approaches such as explainability techniques, such as Grad-CAM, have shown promise in making model decisions more transparent and understandable to clinicians.

In terms of future work, there are several discoveries that researchers can further pursue to improve outcome detection and diagnosis of sickle cell anemia. These include the use of ensemble learning, investigation of larger-scale and more diverse datasets, concerns about model interpretability and risks of overfitting, exploration of more advanced data augmentation techniques, development of more robust and interpretable models, and large-scale clinical validation to assess the effectiveness of diagnostic systems in real-world scenarios.

Furthermore, it is essential to consider the implementation of these technologies in resource-constrained healthcare systems, ensuring their accessibility and utility in diverse clinical contexts.

## Supporting information

**S1 File. PRISMA checklist.**
(DOCX)

**S2 File. List of articles included in the review and list of all titles returned (including repeats).**
(DOCX)

**S3 File. Risk of bias.**
(DOCX)

**S1 Table. Table of all studies identified in the literature search.**
(XLSX)

## Author Contributions

**Conceptualization:** Tiago Fernandes Machado, Marilda de Souza Gonçalves, Cynara Gomes Barbosa, Marcos Ennes Barreto.

**Investigation:** Tiago Fernandes Machado.

**Methodology:** Tiago Fernandes Machado.

**Supervision:** Cynara Gomes Barbosa, Marcos Ennes Barreto.

**Writing – original draft:** Tiago Fernandes Machado.

**Writing – review & editing:** Tiago Fernandes Machado, Francisco das Chagas Barros Neto, Marilda de Souza Gonçalves, Cynara Gomes Barbosa, Marcos Ennes Barreto.

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
