## [Decision Letter · Decision Letter 0]

28 Aug 2024

PONE-D-24-26275Exploring the use of Machine Learning algorithms on Sickle Cell Disease patient data: A systematic reviewPLOS ONE

Dear Dr. Machado,

Thank you for submitting your manuscript to PLOS ONE. After careful consideration, we feel that it has merit but does not fully meet PLOS ONE’s publication criteria as it currently stands. Therefore, we invite you to submit a revised version of the manuscript that addresses the points raised during the review process.

We look forward to receiving your revised manuscript.

Kind regards,

Alireza Sadeghi, M.D., M.P.H.

Academic Editor

PLOS ONE

Journal Requirements:

Additional Editor Comments:

Dear authors,

Thank you for your interesting study. The overall quality of your paper is acceptable. However, there are certain concerns from our esteemed reviewers that need to be addressed.

Also, to improve reproducibility, please provide the exact syntax used for each database to search for records.

Regards,

Alireza Sadeghi

Reviewers' comments:

Reviewer's Responses to Questions

**Comments to the Author**

1. Is the manuscript technically sound, and do the data support the conclusions?

Reviewer #1: Yes

Reviewer #2: Partly

2. Has the statistical analysis been performed appropriately and rigorously? 

Reviewer #1: N/A

Reviewer #2: N/A

3. Have the authors made all data underlying the findings in their manuscript fully available?

Reviewer #1: Yes

Reviewer #2: Yes

4. Is the manuscript presented in an intelligible fashion and written in standard English?

Reviewer #1: Yes

Reviewer #2: No

5. Review Comments to the Author

Reviewer #1: Machado et al. present a systematic review that summarizes a large number of projects using machine learning to diagnose or monitor sickle cell disease. Many of these machine learning methods were successful at addressing their goals and achieved accuracy above 90%.

The materials and methods are clearly described, with Figure 1 showing how the initial set of potential articles was filtered down to 29 articles for systematic review.

The manuscript has an acronym problem. Acronyms can be useful, after begin defined, to refer to an idea in a more concise manner in subsequent regions of the manuscript. Or they can be completely avoided; the writing will be less concise, but readers don't need to remember any acronyms. *The authors need to decide which approach they want to use.* If they choose to use an acronym, it should be defined only once when the term is first mentioned. One problem example is that the phrase "sickle cell disease (SCD)" occurs 7 places in the paper. Similarly, "Support Vector Machine (SVM)" occurs nine times. Many other acronyms, especially for clinical metrics, are introduced but never used. An extreme case is "total of 14 features, such as weight, mean corpuscular volume (MCV), reticulocyte count (RETIC A), hemoglobin (Hb), reticulocyte count (RETIC%), HbF, platelets (PLTS), lactate dehydrogenase (LDH), neutrophils (white blood cells NEUT), alanine aminotransferase (ALT), bilirubin (BILI), aspartate aminotransferase (AST), bio-blood corporal (BIO)". These acronyms should be dropped to increase readability.

Tables 1 and 2 are important parts of the manuscript, with a brief review of each study. They would be more informative, however, if they included some additional information. At present they include ML methods and metrics of success, but the studies are hard to compare because they're quite heterogeneous. For Table 1, please change the "Method" column to "Method/Data" by adding the type of data (such as "blood smear images") used by each study. That column should also include sample size (number of patients (and/or images)) for each study. Otherwise, one cannot attempt to determine whether the metrics of success are due to the method or the sample size (and if the more successful studies simply used more data). A more minor point: in the "Main Results" Column of Tables 1 and 2, please reduce the number of significant figures to 3; additional significant figures are not very meaningful.

At the end of the "Machine Learning Approaches for Diagnosis", please comment about the overall success of these methods. In other words, respond to questions like, Can machine learning be effectively used to diagnose SCD? Is the automated diagnosis of SCD using machine learning a solved problem? If not, what areas of this challenge still need the most work?

For Table 2, please change the "Method" column to "Method/Data" by adding the type(s) of data (such as "PPG" or "vital signs") used by each study to build the ML classifier.

At the end of the "Machine Learning Approaches for Monitoring", please comment about the overall success of these methods. In other words, respond to questions like, Can machine learning be effectively used to monitor SCD? Is the automated monitoring of SCD using machine learning a solved problem? If not, what areas of this challenge still need the most work?

In the "Discussion" section (lines 430-432), this statement needs explanation and/or correction:

"CNNs, especially when combined with transfer learning techniques, have performed better in diagnosing sickle cell disease than traditional algorithms."

How did the authors make this conclusion? What "traditional algorithms" do the authors mean? The statement should be modified to more clearly and correctly summarize the published studies.

Lines 450-452

This statement needs explanation and/or correction:

"Machine learning techniques can significantly improve sickle cell disease diagnosis and monitoring accuracy and speed, potentially reducing the burden on healthcare systems."

Was current SCD diagnosis and monitoring accuracy and speed discussed in the manuscript? The authors have shown that ML methods can effectively diagnose SCD, but they haven't discussed how this compares with current cost, speed, and reliability of diagnosis.

Lines 487-490

This paragraph can be removed, as it is beyond the scope of the current work. It is factually correct but just not relevant.

Conclusion (last paragraph)

Add a mention of the value of increasing training dataset sizes. This is included in the abstract and should also appear here.

Overall the manuscript provides lots of evidence that ML can be effectively used to diagnose and monitor SCD patients, removing doubts that its apparent success in some studies were unwarranted.

Reviewer #2: This manuscript provides a systemic review of ML tools that were applied to diagnosis/monitoring of sickle cell disease. Authors did conduct extensive research in summarizing available ML literatures, however, I have some concerns that need to be clarified through revision. Please find the attached for more details

6. PLOS authors have the option to publish the peer review history of their article (what does this mean?). If published, this will include your full peer review and any attached files.

Reviewer #1: No

Reviewer #2: No

---

## [Author Response · Author response to Decision Letter 0]

4 Oct 2024

Dear Editor, 

I hope this message finds you well. As requested, we have attached the document titled "Response to Reviewers," which contains our detailed responses to the reviewers' comments and suggestions.

Best regards, Tiago Machado (on behalf of all authors).

---

## [Decision Letter · Decision Letter 1]

23 Oct 2024

Exploring Machine Learning algorithms on Sickle Cell Disease patient data: A systematic review

PONE-D-24-26275R1

Dear Dr. Machado,

We’re pleased to inform you that your manuscript has been judged scientifically suitable for publication and will be formally accepted for publication once it meets all outstanding technical requirements.

Kind regards,

Alireza Sadeghi, M.D., M.P.H.

Academic Editor

PLOS ONE

Additional Editor Comments (optional):

Reviewers' comments:

Reviewer's Responses to Questions

**Comments to the Author**

1. If the authors have adequately addressed your comments raised in a previous round of review and you feel that this manuscript is now acceptable for publication, you may indicate that here to bypass the “Comments to the Author” section, enter your conflict of interest statement in the “Confidential to Editor” section, and submit your "Accept" recommendation.

Reviewer #1: All comments have been addressed

Reviewer #2: All comments have been addressed

2. Is the manuscript technically sound, and do the data support the conclusions?

Reviewer #1: Yes

Reviewer #2: Yes

3. Has the statistical analysis been performed appropriately and rigorously? 

Reviewer #1: Yes

Reviewer #2: Yes

4. Have the authors made all data underlying the findings in their manuscript fully available?

Reviewer #1: Yes

Reviewer #2: Yes

5. Is the manuscript presented in an intelligible fashion and written in standard English?

Reviewer #1: Yes

Reviewer #2: Yes

6. Review Comments to the Author

Reviewer #1: (No Response)

Reviewer #2: All my comments are well address. The clarity of ML has been improved substantially. The manuscript is acceptable if other reviewers also agree.

7. PLOS authors have the option to publish the peer review history of their article (what does this mean?). If published, this will include your full peer review and any attached files.

Reviewer #1: No

Reviewer #2: No

---

## [Editor Report · Acceptance letter]

1 Nov 2024

PONE-D-24-26275R1 

PLOS ONE

Dear Dr. Machado, 

I'm pleased to inform you that your manuscript has been deemed suitable for publication in PLOS ONE. Congratulations! Your manuscript is now being handed over to our production team.

Kind regards, 

on behalf of

Dr. Alireza Sadeghi 

Academic Editor

PLOS ONE